# Chronic Critical Illness and PICS Nutritional Strategies

**DOI:** 10.3390/jcm10112294

**Published:** 2021-05-25

**Authors:** Martin D. Rosenthal, Erin L. Vanzant, Frederick A. Moore

**Affiliations:** Division of Acute Care Surgery and the Sepsis and Critical Illness Research Center, University of Florida College of Medicine, Gainesville, FL 32610, USA; erin.vanzant@surgery.ufl.edu (E.L.V.); Frederick.Moore@surgery.ufl.edu (F.A.M.)

**Keywords:** persistent inflammation immunosuppression catabolism syndrome, chronic critical illness, malnutrition, PICS, immunonutrition, high protein

## Abstract

The nutritional hallmark of chronic critical illness (CCI) after sepsis is persistent inflammation, immunosuppression, and catabolism syndrome (PICS), which results in global resistance to the anabolic effect of nutritional supplements. This ultimately leaves these patients in a downward phenotypic spiral characterized by cachexia with profound weakness, decreased capacity for rehabilitation, and immunosuppression with the propensity for sepsis recidivism. The persistent catabolism is driven by a pathologic low-grade inflammation with the inability to return to homeostasis and by ongoing increased energy expenditure. Better critical care support systems and advances in technology have led to increased intensive care unit (ICU) survival, but CCI due to PICS with poor long-term outcomes has emerged as a frequent phenotype among ICU sepsis survivors. Unfortunately, therapies to mitigate or reverse PICS-CCI are limited, and recent evidence supports that these patients fail to respond to early ICU evidence-based nutrition protocols. A lack of randomized controlled trials has limited strong recommendations for nutrition adjuncts in these patients. However, based on experience in other conditions characterized by a similar phenotype, immunonutrients aimed at counteracting inflammation, immunosuppression, and catabolism may be important for improving outcomes in PICS-CCI patients. This manuscript intends to review several immunonutrients as adjunctive therapies in treating PICS-CCI.

## 1. Introduction

Advances in intensive care unit (ICU) technology, support systems, care bundles, and evidence-based protocols have improved ICU survival by largely preventing early deaths [1]. Girard and Raffin in 1985 first described the transition from acute critical illness to chronic critical illness (CCI) as the need for acutely ill patients to require ongoing chronic support in the ICU setting [2]. As patients require ongoing care throughout chronic phases, the ICU setting has been extended to include care provided at long-term acute care facilities (LTACs). The clinical definition of CCI is not well established and quite variable. Most early descriptions involved patients requiring prolonged mechanical ventilation and discharge to LTACs for ventilator weaning. A variety of descriptive terms have been used in these reports such as the myopathy of critical illness, neuropathy of critical illness, and most recently the post intensive care unit syndrome. Kahn and Iwashyna et al. reported using a survey of inpatient databases that 5–8% of all ICU admissions (medical and surgical) develop CCI, accounting for greater than 380,000 annual cases, 107,000 in-hospital deaths, and over USD 26 billion in health care costs [3,4]. However, this relatively small patient population consumes over 30% of ICU resources [4,5]. This growing body of literature now has been extended to include many different types of ICU patients; however, a major omission is the limited amount of information and ongoing research related to the underlying pathobiology of CCI. The University of Florida (UF) Sepsis Critical illness Research Center (SCIRC) has focused its efforts over the past decade on characterizing CCI in surgical ICU patients who share a similar response after trauma, burns, and sepsis. When this response becomes excessive and dysregulated, it causes an injurious systemic inflammatory response syndrome (SIRS, with multiple organ dysfunctions), an immunosuppressive compensatory anti-inflammatory response syndrome (CARS, with secondary infections), and catabolic stress metabolism (with loss of lean body mass) [6]. Recent studies showed that these responses occur simultaneously and that it is the failure to return to homeostasis that characterizes the CCI clinical trajectory [7,8,9]. In a 2012 review article, UF SCIRC investigators coined the term persistent inflammation, immunosuppression, and catabolism syndrome (PICS) to provide a mechanistic framework in which to study CCI in surgical ICU patients [10]. From 2014 to 2019, they enrolled 363 septic patients who they followed longitudinally for 1 year in a NIH-funded prospective study to define the epidemiology, dysregulated immunity, and long-term outcomes of new onset sepsis in surgical ICU patients [11]. Patients were categorized into three clinical trajectories: (1) early death (within 14 days), (2) rapid recovery (RAP, ICU stay < 14 days), and (3) CCI (≥14 days of ICU stay with ongoing organ dysfunction based on SOFA score). It was observed that early mortality was surprisingly low (only 4%) and that 63% of patients experienced RAP. However, one third of survivors progressed into CCI. Unfortunately, most of the CCI patients were discharged to non-home destinations with severe functional and cognitive disabilities from which they do not recover, and 40% were dead at 1 year [12,13,14]. These poor long-term outcomes after sepsis are consistent with other recent reports [15,16,17,18,19]. CCI patients tend to be discharged frail and to suffer from significant pain, dyspnea, psychological distress, fatigue, and delirium related to their protracted ICU and hospital stay [20,21,22,23,24,25]. CCI patients who survive have poor quality of life, with many suffering from depression, cognitive impairments, complex physiologic abnormalities, and chronic organ dysfunction from which they rarely recover [24,26,27,28,29,30,31,32,33].

In 2020, Rosenthal et al., published an analysis of CCI patients from the UF SCIRC sepsis database in which it was found that, despite receiving adequate macronutrients starting early in the ICU, CCI patients did not respond the way RAP patients did. Rather, they experienced a persistent acute phase response (with high C reactive protein and negative acute phase reactants, as depicted by low albumin levels) and failed to become anabolic [34]. These data provide the rationale for additional nutritional adjuncts in PICS-CCI patients. This manuscript reviews specific nutritional adjuncts that likely are beneficial for patients with chronic diseases that experience a similar phenotype of persistent inflammation, anabolic resistance, and cachexia. These are discussed in the following sections: Protein Supplements, Specific Amino Acids, Omega 3 Fatty Acids (FA) and Specialized Pro-Resolving Mediator (SPMs) Supplementation, Probiotics, and Anabolic Agents.

## 2. Protein Supplements

Over the past few decades, guideline recommendations for ICU protein supplementation have increased from 0.8 g/kg/day up to as high as 2 g/kg/day (2.5 g/kg/day in renal replacement therapy) in an attempt to offset catabolism and anabolic resistance [35]. The emphasis is no longer on positive caloric balance but rather on providing specific macronutrients that drive clinical outcomes for ICU patients. During periods of physiologic stress, the body mobilizes and then catabolizes large amounts of protein from muscle that functionally debilitates patients [36]. After trauma and sepsis, resting energy expenditure peaks at around 5 days but can continue for up to 12 days, losing up to 16% of total body protein [37,38,39]. Weijs et al. showed that prescribing early high protein had survival benefit in ICU patients yet overfeeding was linked to increased mortality [40,41]. Allingstrup et al. added that a higher protein prescription (>1.46 g/kg/day vs. 1.06 or 0.79 g/kg/day) and specific amino acids supplementation (either Viamin 18 or Glavamin, Fresenius Kabi, Germany) was again associated with a lower mortality [42]. Compher and Heyland et al. reported that increased protein delivery had a significant survival benefit in nutritionally high-risk patients based off the Nutrition Risk in the Critically Ill (NUTRIC) > 5 but not significantly so in nutritionally low-risk patients [43]. In 2013, Wolfe and Deutz et al. described an “anabolic response”, where higher protein supplementation suppressed endogenous protein catabolism, and that this anabolic response was dose dependent to the higher amount of protein provided [44]. All of these studies support protein as the single most important macronutrient driving clinical outcomes in ICU patients, which likely impacts patients throughout their length of hospital stay providing benefit to PICS-CCI patients.

Adding to the rationale for higher protein supplementation in PICS-CCI was another recent publication from a Protein Summit that recommended protein prescription in the range of 1.2–2.5 g/kg/day in the ICU setting to optimize nutrition, to preserve muscle mass, and to decrease mortality [45]. Finally, Rondanelli et al. identified that the combination of exercise and protein supplementation with vitamin D and an improved omega 6 to omega 3 FA ratio (either omega 3 supplementation directly or consumption of fish > 4 times/week) maintained lean muscle mass in elderly patients [46]. This again has implications for PICS-CCI patients to combat catabolism by potentially feeding them with increasing doses of protein to improve long-term outcomes.

## 3. Specific Amino Acids

Arginine is a conditional amino acid with a wide range of bioactive impact but under states of physiologic stress and ongoing inflammation becomes depleted. Barbul and colleagues were early champions of arginine metabolism and reported its importance in wound healing and as an immunologic regulator [47,48,49,50,51,52]. Arginine serves as a substrate for nitric oxide (NO) production, causing vasodilation in tissue to enhance the delivery of oxygen and nutrients to healing wounds [51,52,53,54,55,56]. Additionally, arginine serves as an intra-cellular substrate for NO production in macrophages to improve bactericidal activity as well as improves T-cell function, proliferation, and maturation to reverse immunosuppression among ICU patients [47,48,49,55,57,58,59]. While arginine use in acute sepsis remains controversial, we have shown that the persistent expansion of myeloid-derived suppressor cells (MDSCs) characterizes PICS-CCI and is predictive of secondary infections and late mortality. MDSCs upregulate arginase-1, causing an arginine deficiency, which drives immunosuppression by affecting lymphocyte proliferation and maturation and renders the zeta-chain of the T-cell receptor (TCR) dysfunctional, thus causing T-cell incompetency [57,60,61,62,63,64,65,66]. PICS-CCI patients are typically well outside their sentinel septic event that results in inflammation and immunosuppression, thus consequently the unfounded controversy concerning arginine aggravating an acutely septic state is irrelevant. Therefore, providing this amino acid to PICS-CCI patients makes logical sense to promote recovery and healing and to potentially reverse some of the immunosuppression that this population endures.

Branched-chain amino acids (BCAAs: leucine, isoleucine, and valine) are another group of amino acids that, when supplemented, have demonstrated decreased muscle catabolism and hypertrophic muscle growth through increased protein synthesis [67]. Frank Cerra championed an idea of “septic auto-cannibalism” that occurred in multiple organ failure (MOF) patients despite standard parenteral nutrition (PN) and recommended using BCAA to combat the ongoing ICU muscle breakdown [68]. In a prospective randomized trial, Cerra reported that the use of BCAA-fortified PN in surgical patients improved visceral protein markers and nitrogen balance as well as absolute lymphocyte count, which is one of the laboratory parameters of PICS-CCI. As these BCAA formulas were expensive, clinical use waned until recent studies linked leucine stimulation of the Mammalian Target of Rapamycin (mTOR) pathway with increases in protein synthesis (anabolic) and inhibition of proteosomal protein breakdown (anti-catabolic) [40,69]. During PICS-CCI, leucine and potentially even beta-hydroxy-beta-methylbutyrate (HMB: a metabolite of leucine) may help decrease and potentially even reverse the catabolic nature of their pathophysiologic state [70]. It is well known that critically ill patients lose lean muscle mass at an accelerated rate while being bed bound, as described above [71]. Prolonged ICU stays, associated with the chronicity of PICS-CCI patients, only adds to the persistent catabolism, giving rise to ICU-acquired weakness and a PICS phenotype with poor rehab potential [72]. Leucine or HMB supplementation hopefully preserves or increases muscle mass and strength to improve patients rehab potential and to restore some semblance of baseline function once released from the ICU. Leucine works in concert with arginine to stimulate mTOR for its required anabolic, muscle hypertrophy effect [73]. Ultimately, in supplementing these two amino acids, the goal is to allow the synergistic effect of arginine and leucine to promote an anabolic recovery sooner than expected in PICS-CCI patients.

Glutamine supplementation may prove to be important as its pluripotent ability to serve as an antioxidant, in gluconeogenesis, and to enhance immune function is lost during times of great stress. Skeletal muscle generates the majority of endogenous glutamine, but during major catabolic insults, demand for glutamine becomes far greater than supply. This phenomenon results in glutamine becoming a conditional amino acid. Glutamine serves as the primary oxidative fuel for rapidly dividing tissues, such as small bowel mucosa, proliferating lymphocytes, and macrophages [74]. Glutamine also serves in intermediary metabolism, including the maintenance of acid–base status, as a precursor of urinary ammonia, and in nitrogen transfer for the biosynthesis of nucleotides, amino sugars, arginine, glutathione (an antioxidant), and glucosamine [75,76]. During periods of stress, glutamine can provide the carbon skeleton for gluconeogenesis and is the primary substrate for renal gluconeogenesis [77]. Some studies have suggested that glutamine supplementation boosts cell-mediated immunity through enhanced proliferation of lymphocytes while simultaneously suppressing systemic inflammation through cessation of pro-inflammatory cytokines [78]. These beneficial effects on immune function and inflammation are believed to contribute to the lower rates of infection and inflammatory complications in critically ill patients who receive glutamine supplementation [79]. Though PICS-CCI patients are typically removed from their inciting pathophysiologic insult, they remain immunosuppressed and catabolic and have a high rate of sepsis recidivism. By providing the aforementioned amino acids, the hope is to fortify PICS-CCI patients with a specific amino acid that provides substantial non-caloric benefit.

## 4. Omega 3 Fatty Acids (FA) and Specialized Pro-Resolving Mediator (SPMs) Supplementation

Lipids remain a crucial macronutrient and mainstay nutritional therapy for the critically ill and surgical populations, but controversy still exists regarding the digestion, absorption, and oxidation of lipids in hyperdynamic states [80]. While uncertainty may exist on which lipid formulation to deliver, there is no debate regarding the need to meet the essential FA and cellular oxidative requirements. The beneficial anti-inflammatory effects of omega 3 FAs (primarily eicosapentaenoic acid (EPA) and docosahexaenoic acid (DHA)) have been well documented in several chronic inflammatory diseases, including rheumatoid arthritis, Crohn’s disease, ulcerative colitis, lupus, multiple sclerosis, and asthma [81,82,83,84,85]. Recent evidence from an updated meta-analysis by Pradelli et al. suggests a similar benefit with omega 3 supplementation. This meta-analysis included close to 50 randomized control trails (most are surgical patients) and found that parenteral omega 3 FAs decreased hospital length of stay and ICU length of stay, overall infectious complications, and septic events and was financially cost saving for the duration of ICU patients [86,87,88]. It has also been reported in numerous human randomized clinical trials that the use of omega 3 FAs (EPA/DHA) can partially attenuate the hypermetabolic response to critical illness, can minimize lean muscle loss, can inhibit oxidative injury, and can improve outcomes by modulating the synthesis of pro-inflammatory and by providing anti-inflammatory mediators [80,89,90,91]. It appears that 2 g/day of EPA and DHA orally or 0.2 g/kg/day parenterally are effective at eliciting the desired anti-inflammatory actions; however, few studies specifically looked at dose to effect outcomes [92,93,94].

An advanced understanding of the bioactive nature of omega 3 FAs (EPA/DHA) has shown us that there is an even greater endogenous byproduct of lipid metabolism called Specialized Pro-Resolving Mediator (SPMs) [95]. These highly active SPMs are classified into three species called resolvins, maresins, and protectins [96]. They are found in nano- and picomolar concentrations and help to resolve inflammation [97,98]. Serhan and colleagues have shown that SPMs decrease inflammation by halting leukocyte infiltration, activation, and enhanced innate immune function of clearing debris, bacteria, and apoptotic cells [96,97]. SPMs hypothetically can be impactful in attenuating the dysfunctional inflammatory response and immunosuppression and in decreasing catabolism through the preservation of endogenous energy no longer diverted to the chronic systemic hyper-inflamed state observed in PICS-CCI.

## 5. Probiotics

ICU care, as advanced as it may seem, has limitations, which has generated research interests for a better understanding of what a healthy, diverse microbiome can provide a host during critical illness. Since 1907, when the Nobel Laureate Elie Metchnikoff first described the concept of probiotics, the overarching opinion has been that probiotics are safe and have a role in treating gastrointestinal diseases, but the use has not been supported by the literature [99]. Insight into how the stresses in ICU care, critical illness, and surgery all negatively impact the microbiome has led research teams to identify that these stresses give rise to virulent organisms collectively called the pathobiome [100,101,102,103,104,105,106,107]. Various strategies have been reported to reduce the burden of the pathobiome, but none seem as promising as probiotics.

Supplementing the critically ill and post-surgical patients with both pre- and probiotics has proven beneficial in many settings, but evidence lacks as to which specific species are superior to improve the dysbiosis and can they be beneficial to PICS-CCI? Despite the amounting research that has shown benefit, supplementing probiotics is still not considered the standard of care in all ICU settings. Nevertheless, the idea is that ‘bioecological control’ has blossomed in critically illness and polytrauma patients: to supply viable beneficial bacteria or a substrate that enhances these specific beneficial bacteria instead of trying to eliminate the pathogen [108,109]. Could this mean that, by fortifying the gut microbiota through probiotics, we could alter or enhance the recovery of PICS-CCI patients?

Anton et al. performed a meta-analysis to identify nutritional and pharmacologic interventions targeting chronic low-grade inflammation in older adults. In this analysis, he identified probiotics as having the largest impact on decreasing the serum biomarkers of inflammation (IL-6 and CRP). Angiotensin II receptor blockers and omega 3 fatty acids also significantly reduced these biomarkers but to a lesser extent compared to probiotics [110]. Other chronic states of inflammation similar to PICS-CCI are associated with inflammation, catabolism, and anabolic resistance, leading to frailty and poor long-term outcomes. It is this “inflammaging” described by Franceschi that contributes to age-related decline in functional status and increases in morbidity and mortality [111]. Perhaps a better understanding of how our gut microbiome cross talks with the body in health and illness will elucidate which specific probiotic can be most beneficial for PICS-CCI?

## 6. Anabolic Agents

Anabolic and anti-catabolic agents may be beneficial in mitigating persistent muscle breakdown, a pillar of PICS-CCI. Herndon and colleagues described numerous adjuncts to mitigate catabolism in the pediatric burn population that may be of value in the PICS-CCI population, including (a) growth hormone [112], (b) intensive insulin therapy [113,114], (c) oxandralone [115,116], (d) propranolol [117], and (e) exercise programs [118]. Growth hormone and, arguably, each of these adjunctive therapies have a net response to be “potent anabolic agent and salutary modulators of posttraumatic metabolic responses” [112]. These therapies preserve lean muscle mass, increase strength, enhance bone mineralization, and attenuate the hypermetabolic response to burn injury, leading to quicker recovery [113,114,119].

## 7. Conclusions

PICS-CCI is the high cost of ICU survival and the new, dominant chronic MOF phenotype in ICUs and LTACs. The underlying pathobiology contributing to this increasingly common CCI phenotype are being unraveled. We now know that evidence-based, gold standard nutritional protocols fail to have much of an effect on the PICS-CCI phenotype as this patient population continues to exhibit a cachexia-like response to our nutritional strategies. Commencing therapeutic strategies early in critical illness may mitigate progression into CCI. When PICS-CCI is established, combatting the pillars will likely require multiple modalities. Nutritional strategies such as the use of various immunonutrients are forthcoming and are one modality to hopefully alter the clinical trajectory of PICS-CCI. Among these strategies, the ones that hold the highest potential are protein concentration ≈2 g/kg/d, specific immuno-amino acids, omega 3 fatty acids, specialized pro-resolving mediators, probiotics, and anabolic agents.

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
