# Peer review of "Chronic Critical Illness and PICS Nutritional Strategies"

_jcm, 2021, doi:10.3390/jcm10112294_

Round 1

Reviewer 1 Report

The review is generally well structured. This is not really review of nutritional strategies in critical illness, it is more a review of the effects of different supplements on critical illness. The problem is - it details many effects on biomarkers, but very little in the way of clinical outcomes. Indeed, there is a good deal of conjecture, even when the evidence contradicts the authors view. e.g. 

"Each of these adjunctive therapies have a net response to be “potent anabolic agent and salutary modulators of posttraumatic metabolic responses.”These therapies preserve lean muscle mass, increase strength, enhance bone mineralization, and attenuate the hypermetabolic response to burn injury; leading to a quicker recovery. They do not lead to faster recovery and the references which they reference do not support this.

There is a good deal of self-referencing

The authors place a good deal of weight on expert opinion rather than solid science.

P2 L51 Should read: In a 2012 review.....

P3 L110-11 - this sentence is missing a word - around the word "potential"

Author Response

The review is generally well structured. This is not really review of nutritional strategies in critical illness, it is more a review of the effects of different supplements on critical illness. The problem is - it details many effects on biomarkers, but very little in the way of clinical outcomes. Indeed, there is a good deal of conjecture, even when the evidence contradicts the authors view. e.g. First I wanted to thank the reviewer for their comments and timely analysis of this article.  In reference to the above comments I agree this is not an overall review of nutritional supplements, therapies, or adjuncts in acute critical illness but rather for chronic critical illness and PICS.  There is amble level one evidence and major societal recommendations from SCCM/ASPEN for critical care nutrition.  Little is known about nutritional adjuncts for chronic critical illness and none for Persistent Inflammation Immunosuppression and Catabolic Syndrome as this last phenotype has only been discussed within the past 10 years. To discuss all of critical care nutrition is outside the scope of this review as it only provides a review of how to approach different pathophysiologic states in the construct of the 3 pillars of PICS-CCI.

"Each of these adjunctive therapies have a net response to be “potent anabolic agent and salutary modulators of posttraumatic metabolic responses.”These therapies preserve lean muscle mass, increase strength, enhance bone mineralization, and attenuate the hypermetabolic response to burn injury; leading to a quicker recovery. They do not lead to faster recovery and the references which they reference do not support this."

In reference to this comment above I have changed the structure to read "Growth hormone, and arguably each of these adjunctive therapies, have a net response to be “potent anabolic agent and salutary modulators of posttraumatic metabolic responses.”[107 This work was done by David Herndon's group in Galveston TX and he has shown largely in pediatric burn population that each of these adjuncts have the ability to promote catabolic resolution quicker through an anabolic response which could be important for our PICS-CCI cohort. 

 There is a good deal of self-referencing...There indeed is some nepotism when discussing the current literature on PICS-CCI as the phenotype was first described and published on by this institution with the largest series of publications on PICS-CCI discussing the epidemiology, immune response, MDSCs, and PICS-CCI in both sepsis and trauma. This article stands as one of several reviews this institution has been invited to submit for a special edition on PICS-CCI.

As far as the bottom 2 comments I have corrected those thanks for catching these mistakes.

Reviewer 2 Report

In this manuscript Rosenthal et al. review the current evidence regarding (immuno)nutriton in chronic critical illness (CCI) and presistent inflammation, immunosuppression and catabolism syndrome (PICS). They specify about protein, specific amino acids, fatty acids, probiotics and anabolic agents.

I read this narrative review with great interest. The manuscript is well written, but has some limitations:

  • The first part of the introduction cites a former review of the same group. Is there original literature rather than a review?
  • Exept for protein there are no clear recommendations for tailoring nutrition in daily caring for PICS-CCI patients. Maybe there is too few data? What information/recommendation should the reader gain after this reading?
  • The intention of the article in reviewing specific nutritional adjuncts is stated only at the end of the introduction. Why not in the abstract?
  • You discuss arginine, BCAAs and HMB. Is there a reason for choosing exactly these amino acids? Why not eg. taurine or others?
  • The conclusion is vague and very short given the profoundness of the sections 2, 3, 4 and 5. I think outlining the main findings of these sections could aid in understanding of such a complex topic, even for experienced ICU physicians.
  • Why not state "(narrative) review" in the title?

Minor comments:

  • Line 121: [...] in acute sepsis is remains [...], remove one verb
  • Line 124: [...] MDCSs upregulate [...], mixed up letters
  • Line 192: [...] laureate: Elie [...], spare colon

Author Response

  • I wanted to first thank the reviewer for taking their time in reviewing this manuscript and providing feedback. 
  • The first part of the introduction cites a former review of the same group. Is there original literature rather than a review? Please see bellows answer to the second question for comments as they are similar
  • Exept for protein there are no clear recommendations for tailoring nutrition in daily caring for PICS-CCI patients. Maybe there is too few data? What information/recommendation should the reader gain after this reading? Ultimately your comments to this effect are on point.  You have recognized that there is little to no data.  The review is on PICS-CCI for which their is scant literature on chronic critical illness other than to say its recognized phenotype.  Providing any supplements, let alone nutritional, is typically made as inferences from other bodies of literature. When dealing with PICS, as this phenotype has only recently been proposed in the past decade, has no randomized control trials looking specifically at this patient population. 
  • The intention of the article in reviewing specific nutritional adjuncts is stated only at the end of the introduction. Why not in the abstract? I have added it to the abstract
  • You discuss arginine, BCAAs and HMB. Is there a reason for choosing exactly these amino acids? Why not eg. taurine or others? Thank you for this feed back I have added a section on glutamine. Specifically about taurine my understanding is that there is great enthusiasm building for this amino acid as it is recognized that sulfur containing amino acids are low during sepsis and tied to poor outcomes when deficient but randomized control trials are lacking.  I have only found 1 RCT out of Cairo, we don't know the best dose, we still don't know toxicity levels in humans, but understand from a hypothetic point it serve to protect ischemic-reperfusion injury as an antioxidant.
  • The conclusion is vague and very short given the profoundness of the sections 2, 3, 4 and 5. I think outlining the main findings of these sections could aid in understanding of such a complex topic, even for experienced ICU physicians. Thank you for this comment. I have added and the entered this statement to help. 

    Nutritional strategies such as the use of varies immunonutrients are forthcoming and are one modality to hopefully alter the clinical trajectory of PICS-CCI. Among these strategies that hold the highest potential are protein concentration ~2g/kg/d, specific immuno-amino acids, omega 3 fatty acids, specialized pro-resolving mediators, probiotics, and anabolic agents.

  • Why not state "(narrative) review" in the title? My understanding is there is no section for the journal to provide narrative review just reviews

The remain critiques have been addressed. Thanks for this thoughtful review.

Round 2

Reviewer 1 Report

There hasn't really been a major revision to this manuscript. The authors have added a paragraph and corrected some words - it doesn't really alter the substance of the article. Whilst I agree that nutrition is a major part of the SCCm guidance - I am not sure their article links the biochemical with the clinical - which was the point I was making. There are no nutritional strategies (as their title suggests) - just a set of statments from the literature - mainly regarding biochemical outcomes and not clinical outcomes.

Author Response

I thank you again for your review, but disagree with your comments.  If you look at the section of each of these strategies you will find that it is tied to clinical outcomes.  The evidence for each of these nutritional strategies often times promotes anabolism whether directly such as the case of protein or indirectly through decreasing energy expenditure through inflammation (as in omega 3 lipids and SPMs).  So despite the fact that dosing may not be specified becuase often times we may not know the appropriate dose like with SPM, the main recommendation is to use these nutritional strategies to fortify the patient and provide clinical benefit to attack the 3 pillar of persistent inflammation, immunosuppression, and catabolic syndrome.

Reviewer 2 Report

Thank you for replying to all questions and adding the glutamine part as well as the final statement on nutritional strategies.

Author Response

Thank you kindly for this response and review!

This manuscript is a resubmission of an earlier submission. The following is a list of the peer review reports and author responses from that submission.